



# Bio-optical properties of cyanobacterium *Nodularia spumigena*

Shungudzemwoyo P. Garaba [1,*], Michelle Albinus[1], Guido Bonthond [2], Sabine Flöder [3], Mario L. M. Miranda [4,5], Sven Rohde [6], Joanne Y. L. Yong [3] and Jochen Wollschläger [1]

[1] Marine Sensor Systems Group, Center for Marine Sensors –- Institute for Chemistry and Biology of the Marine Environment, Carl von Ossietzky University of Oldenburg, Schleusenstraße 1, 26382 Wilhelmshaven, Germany
[2] Environmental Biochemistry Group, Institute for Chemistry and Biology of the Marine Environment, Carl von Ossietzky University of Oldenburg, Schleusenstraße 1, 26382 Wilhelmshaven, Germany
[3] Plankton Ecology Group, Institute for Chemistry and Biology of the Marine Environment, Carl von Ossietzky University of Oldenburg, Schleusenstraße 1, 26382 Wilhelmshaven, Germany
[4] Laboratorio de la calidad del agua y Aire, Universidad de Panamá, 0824, Panamá
[5] Sistema Nacional de Investigación, Secretaría Nacional de Ciencia y Tecnologías, Panamá.
[6] Environmental Biochemistry Group, Institute for Chemistry and Biology of the Marine Environment, Carl von Ossietzky University of Oldenburg, Schleusenstraße 1, 26382 Wilhelmshaven, Germany

*Correspondence to*: Shungudzemwoyo P. Garaba (shungu.garaba@uni-oldenburg.de)

## Abstract

In the last century an increasing number of extreme weather events have been experienced across the globe. These events have also been linked to changes in water quality especially due to heavy rains, flooding, or droughts. In terms of blue economic activities, one major threat are harmful algal bloom events that tend to be widespread and can last up to several days. We present and discuss advanced measurements of a bloom involving the cyanobacterium *Nodularia spumigena* conducted by hyperspectral optical technologies through experiments-of-opportunity. Absorption coefficients, absorbance and fluorescence were measured in the laboratory and these are available via (https://doi.org/10.4121/21610995.v1 (Wollschläger et al., 2022), https://doi.org/10.4121/21822051.v1 (Miranda et al., 2023) and https://doi.org/10.4121/21904632.v1 (Miranda and Garaba, 2023). Data used to derive the above-water reflectance is available via https://doi.org/10.4121/21814977.v1 (Garaba, 2023) and https://doi.org/10.4121/21814773.v1 (Garaba and Albinus, 2023). Additionally, hyperspectral fluorescence measurements of the dissolved compounds in the water were done. The hyperspectral measurements were conducted over a wide spectrum (200 – 2500 nm). Identification of the cyanobacterium was completed by visual analyses under a microscope. Diagnostic optical features were determined using robust statistical techniques. Water clarity was inferred from Secchi disk measurements https://doi.org/10.1594/PANGAEA.951239 (Garaba and Albinus, 2022). Full sequences were obtained of the 16S rRNA and rbcL genes revealing a very strong match to *Nodularia spumigena*, data available via GenBank https://www.ncbi.nlm.nih.gov/nuccore/OP918142/ (Garaba and Bonthond, 2022b) and https://www.ncbi.nlm.nih.gov/nuccore/OP925098 (Garaba and Bonthond, 2022a). The chlorophyll-a and phycocyanin levels determined are in open-access https://doi.org/10.4121/21792665.v1 (Rohde et al., 2023). Our experiments-of-opportunity echo the importance of sustainable, simplified, coordinated and continuous water quality monitoring as a way to thrive for the targets



set in the United Nations Sustainable Goals (e.g., Goals 6, 11, 12 and 14) or European Union Framework Directives (e.g. Water, Marine Strategy).

# 1 Introduction

Photosynthetic eukaryotic microalgae as well as cyanobacteria play an important role as primary producers in aquatic
ecosystems. These primary producers form the basis of the aquatic food web, fix carbon, produce oxygen, and are involved in nutrient cycling. However, when certain environmental variables favour the excessive growth and accumulation of any particular species, it becomes an algal bloom. Such an event becomes detrimental (i.e., harmful algal bloom, HAB), when the bloom-forming species produce toxins or have a negative impact on other aquatic organisms due to their inherent sheer biomass in the water (Karlson et al., 2021;Smayda, 1997;Carmichael, 1992;Francis, 1878). Some of the HABs can have damaging
effects to socioeconomic activities including human health, animals, aquaculture, recreational, fishing and tourism industry (Glibert et al., 2005;Hallegraeff et al., 2003;IOCCG, 2021;Mazur and Pliński, 2003;Karlson et al., 2021;Nehring, 1993).

Cyanobacteria are known to cause cyanoHABs in fresh to brackish waters which often manifest as unsightly scum that accumulates on the surface and shores of water bodies. An example is the cyanobacterium *Nodularia spumigena* (*N.*
*spumigena*) found in soil and diverse geographic aquatic environments (Horstmann, 1975;Kahru et al., 1994;Öström, 1976;Mazur and Pliński, 2003;Karlsson et al., 2005;da Silveira et al., 2017). *N. spumigena* is a diazotrophic filamentous cyanobacteria that can form blooms during the northern hemisphere summer when water temperatures rise above ~16 °C under calm weather state with long hours of direct ambient light (Lehtimaki et al., 1997;Wasmund, 1997;Kanoshina et al., 2003;Olofsson et al., 2020). These environmental conditions result in the stratification of the water column and a nutrient
depletion that is conducive for the growth of heterocystous filamentous cyanobacteria species such as *N. spumigena* in the euphotic zone (Karlberg and Wulff, 2013). An increasing number of cyanoHABs are expected in the advent of climate change with extreme weather events producing favourable environmental conditions for blooms (Chapra et al., 2017). Consequently, there is going to be a rise in the demand for near real-time monitoring strategies such as remote sensing capable of offering repeated wide area coverage of all aquatic environments at varying geo-spatial resolution.

Wide area repeated monitoring of *N. spumigena* has been achieved in the Baltic Sea and other geographic locations using multispectral satellite missions in the last few decades (Kahru et al., 1994;Öström, 1976;Kahru and Elmgren, 2014;Galat et al., 1990;Leppänen et al., 1995;Mazur and Pliński, 2003). As some reports have revealed, there are *N. spumigena* strains that can be very harmful and toxic to animals or humans which further stresses the importance of detection and identification of
such related blooms e.g., (Teikari et al., 2018;Nehring, 1993;Sivonen et al., 1989). Furthermore, recent advances in remote sensing technologies have also seen a rising number of laboratory based hyperspectral investigations of HABs to better understand and identify inherent diagnostic spectral features of various algae e.g. *N. spumigena* (Soja-Woźniak et al., 2018).

An interdisciplinary dataset gap in hyperspectral measurements that has been echoed in recent scientific user needs discussions and studies relevant for future satellite missions applicable for aquatic monitoring e.g., (IOCCG, 2021;Bracher et al., 2017;Hu et al., 2022;Castagna et al., 2022). To this end, we report on experiments-of-opportunity where a set of hyperspectral observations were conducted following a *N. spumigena* bloom events in an enclosed water body, Lake Bante in Wilhelmshaven, Germany. These high-quality observations are expected to expand as well as add-on towards open-access spectral reference libraries of hyperspectral inherent and apparent bio-optical measurements that will contribute towards future algorithm development, validation and identification of diagnostic spectral features of blooms resulting from *N. spumigena*.

## 2 Methods and Materials

### 2.1 Field campaign sampling

A dark green dense bloom was visually observed on the Lake Bante also known as Banter See in Wilhelmshaven, Germany, on 16 August 2021 (**Figure 1**). Lake Bante is a former harbour basin cut off from the rest of the port exhibiting brackish to freshwater conditions extending ~2.5 × 0.6 km with a maximum depth of 22 m. From observation and general knowledge it is known that the apparent colour of the water in the lake is relatively light greenish. The area of interest is nearby a walking path that is frequented by locals and a variety of birds including ducks. In this study, an absence of these ducks was one of the primary indicators that triggered a further investigation of the easily accessible part of Lake Bante, leading to our discovery of the bloom and hence our experiment-of-opportunity. Sampling was conducted using a measuring cup and two Schott Duran 1000 mL bottles that had been pre-rinsed three times with the sample water. Samples of water from areas that could be considered in bloom and non-bloom conditions, respectively, were collected close to each other. We also visually distinguished these waters based on the apparent colour and density of bloom material. Laboratory measurements were conducted immediately after sampling, therefore no storage in the fridge or dark was considered necessary.

A similar bloom appeared in Lake Bante and was surveyed on 25 August 2022 from 10:00 to 12:00 UTC and sampling was conducted at 14 stations from a small electric powered boat (**Figure 1b**). The water samples were collected following the steps used in the August 2021 campaign, but also additional parameters like geolocation, time, salinity and temperature information for each station was gathered using a SonTek CastAway CTD. Storage of the samples was done in a fridge at 4 °C until filtration or further analyses. Calibrated data was retrieved from the CTD via bluetooth connection in the CastAway™-CTD software version 1.5. Water transparency was inferred from Secchi disk depth measurements. The Secchi disk had a diameter of 30 cm with four equal quadrants in alternating black and white colours.



## 2.2 Laboratory imaging

The collected August 2021 bloom water in one of the Schott Duran bottles was made homogenous by shaking before a sample was taken for microscopic analyses. A dilution ratio of 1:2 with sterile seawater was made because the collected sample was relatively thick and viscous due to the high concentration of cyanobacteria filaments. A further dilution of 1:30, 110 µl sample

+ 3190 µl sterile seawater, was completed and left to settle in a Hydro-Bios Apparatebau Utermöhl sedimentation chamber. The samples were examined under a Zeiss Axiovert 10 inverted microscope between 10 – 40 times magnification after the method of Utermöhl (Utermöhl, 1931). The identification of the bloom-causing organism based on cell morphology was completed using literature (Komárek, 2013). Photographs were taken with a Jenoptik ProGres® Gryphax® Kapella microscope camera. Additionally, automated imaging was completed in a Yokogawa Fluid Imaging Technologies Flowcam

8400 system. For the latter, sample preparation involved filtration through a 100 µm mesh to prevent clogging in the system and the analysed volume of the filtrate was 2.5 ml. No further microscopic inspection was conducted for the August 2022 samples as this had been performed for the 2021 campaign.

## 2.3 Deoxyribonucleic acid (DNA) extraction and sequencing

A molecular genetic confirmation of the bloom causing organism was done using the samples from the August 2022 to further

verify the visual taxonomic identification conducted in August 2021. Water samples (10 mL) were filtered using membrane filters Whatman Nuclepore Track-Etch of 0.2 µm pore size and 47 mm diameter. The filters were frozen at -80 °C until further analysis. After cutting the filters into small fragments, DNA was extracted using the ZYMO Research D6102 fecal/soil microbe kit following the protocol of manufacturer. The 16S rRNA and rbcL genes were amplified in a polymerase chain reaction (PCR) using with the Thermo Scientific Phusion Green Hot Start II High-Fidelity PCR Mastermix and the universal

27F and 1492R primers (5'-AGAGTTTGATCMTGGCTCAG-3' and 5'-GGTTACCTTGTTACGACTT-3') for the 16S rRNA gene and the CX and CW primers (5'-GGCGCAGGTAAGAAAGGGTTTCGTA-3' and 5'-CGTAGCTTCCGGTGGTATCCACGT-3') for the rbcL gene. After an initial denaturation step at 98 ⁰C for 3:00 minutes, the reaction included 30 cycles of 0:30 minutes at 98 ⁰C, 0:30 minutes at 50 ⁰C and 0:30 minutes at 72 ⁰C and a final step of 3:00 minutes at 72 ⁰C. Amplicons were sequenced in forward and reverse direction at Eurofins genomics, Germany. As the

chromatograms of the 16S rRNA gene contained some background signal, cyanobacteria specific primers were designed within the 16S rRNA gene (5'-CCTAGCTTAACTAGGTAAAAAG-3', 5'-TACAAGGCTAGAGTGCG-3' and their reversed complements) and new amplicons were generated and sequenced with the same PCR protocol and program.

## 2.4 Optical measurements and analyses

### 2.4.1 Absorbance

Water samples from the 25 August 2022 survey were filtered through Whatman Nuclepore Track-Etch membranes of 0.2 µm pore size and 47 mm diameter. The filters were frozen and stored at -80 °C. Each frozen filter was put into a separate 20 mL



glass vial before 5 mL 1 × phosphate buffered saline with a pH = 7 was added to begin a pigment concentrations analysis. Sample sonification was completed in an ultrasonic bath filled with crushed ice for four times in 1 minute pulses with 60 second pauses in between. After sonification the glass vial were centrifuged and 3 × 250 µL of the supernatant was transferred onto a 96 microwell plate. A BioTek Instruments Synergy H1 Hybrid Multi-mode microplate reader was used to determine the absorbance of the samples. Phycocyanin concentrations (µg/L) were computed as recommended in the literature (Horváth et al., 2013). The remaining supernatant was decanted and the filters were frozen again at -22 °C. Samples were freeze-dried for a day and then 5 mL of 99.5% ethanol was added to each glass vial. Alcoholic extracts were prepared as explained above and the samples were centrifuged before the supernatant from each sample was measured in the microplate reader. Chlorophyll-a concentrations (mg/L) were determined following the standard protocol (Ritchie, 2008).

**Figure 1** (a) PlanetScope SuperDove satellite-248b true colour RGB composite image captured at 10:07 UTC on 25 August 2022 over Lake Bante in Wilhelmshaven, Germany, (b-c) photographs of the dense bloom as observed around station 13, (d) water samples from 16 August 2021 collected from (53.5093° N, 8.114° E) from the shoreline close to station 1 and (e) radiometric measurements during the 2022 survey.

## 2.4.2 Absorption coefficients

Hyperspectral absorption coefficients in the visible spectrum (400 – 700 nm) were determined directly after sampling using a point-source integrating cavity absorption meter, PSICAM (Kirk, 1997;Röttgers and Doerffer, 2007). The PSICAM had an Illumination Technologies CF1000e halogen lamp as light source and used an Avantes AvaSpec ULS 2048XL-RS-EVO

spectrometer as detector. The instrument was calibrated using a solution of nigrosin with a maximum absorption coefficient of ~0.5 m$^{-1}$. Sartorius Airum® Pro Ultrapure water was used as a reference for both the calibration and sample measurements. For the bloom in 2021, as explained above, the bloom water was very concentrated therefore we diluted the sample at a ratio 1:80 (**Figure 1**, **inset**) and only the total absorption of the water constituents ($a_{tot}$) was measured. The samples of the campaign

in 2022 were analysed for both $a_{tot}$ and coloured dissolved organic material ($a_{cdom}$). For this, the samples were filtered through 0.2 µm pore sized Whatman Nuclepore Track-Etch membrane filters. As the $a_{tot}$ samples were less dense as in 2021, the samples were only diluted at a ratio of 1:10. The $a_{cdom}$ samples remained undiluted. The absorption coefficients of the particulate fraction were obtained by subtraction of $a_{cdom}$ from $a_{tot}$ and denoted as $a_{ph}$, assuming no relevant contribution of non-algae particles.

The absorption coefficient measurements were performed in triplicate meaning the sample was put in and out of the PSICAM three times and an absorption spectrum was obtained. Each measurement was the average of 20 single readings by the spectrometer to minimize noise in the recorded signal which could also contribute to a smoother spectrum. Data processing of absorption spectra incorporates smoothing using loess local regression which is important in case of very low absorption

coefficients. However, we compared unsmoothed and smoothed spectra, and no significant differences were found, which was expected due to the high absorption of the samples. After the measurement of the samples, the respective dilution factor used was considered in the final calculation of the absorption coefficient values.

### 2.4.3 Fluorescence and excitation-emission matrices (EEM) analyses

Bloom water was pre-filtered through Whatman 0.7 µm mesh size GF/F filters and subsequently filtered through Whatman Nuclepore Track-Etch 0.2 µm pore size membranes to remove suspended solids within 72 h after sampling on 16 August 2021. A similar approach was implemented for the August 2022 campaign, but filtration was applied within 2 weeks after field sampling. Sample filtrate from 2021 campaign was diluted with deionized water at a ratio 1:4 to avoid inner filter effect and detector saturation. No dilution was applied for the 2022 samples. A Horiba Aqualog benchtop fluorometer was used to obtain

excitation-emission matrices (EEMs) and absorbance measurements in a 1 cm quartz cuvette containing the sample filtrate. The cuvette had been pre-rinsed three times. Spectra were recorded at 10 nm resolution for both excitation and emission determinations. Excitation was done at 2 nm intervals from the ultraviolet to red spectrum (200 – 600 nm) whilst fluorescence was measured over a wavelength range of 200 nm to 620 nm at 1.617 nm steps. Although emission raw data can be obtained up to 620 nm, we restrict analyses to 600 nm. Laboratory experiments have revealed that beyond 600 nm our Horiba Aqualog

benchtop fluorometer has a relatively low signal-to-noise ratio and sensitivity. A tri-dimensional dataset is generated from the emission, excitation and sample measurement.

Within the scope of this dataset description an additional analysis was performed to highlight the added-value to the scientific community of the obtained EEMs measurements. Using the 2021 dataset only the derived EEMs were corrected using the DrEEM toolbox and further analysed using PARAllel FACtor analysis, PARAFAC (Murphy et al., 2013). The PARAFAC correction removed Raman and Rayleigh scattering components from the measurements followed by a normalization to Raman

units. Multivariable analysis of the tri-dimensional dataset obtained from the EEMs allows for the isolation of principal components associated to specific moieties in the dissolved organic material pool. A dense algae bloom can be characterized, with caveats, by the increment in the metabolic processes responsible for the release of dissolved organic compounds such as polymers, amino acids residues, and decaying cells. Therefore, PARAFAC was utilized to characterize changes in the main composition of surface waters due to *in situ* produced fluorescent dissolved organic matter identifying and isolating fluorescent

components related to microbiological activity. Principal components were validated by a split half analysis in the DrEEM toolbox and a model explaining over 99.5% of the dataset variability was fitted for the analysed samples. PARAFAC was performed using Mathworks Matlab 2017b as detailed in a prior report (Miranda et al., 2020). Inter-comparison of measured EEMs was completed in OpenFlur (Murphy et al., 2014).

**2.4.4 Spectral reflectance and radiance**

Relative hyperspectral reflectance measurements of the undiluted sample were conducted on 12 August 2021 using a Spectral Evolution (SEV) SR-3501 spectroradiometer fitted with an 8° field-of-view lens to collected data interpolated to 1 nm resolution from the ultraviolet (UV, 280 nm) to shortwave infrared (SWIR, 2500 nm) spectrum. Observations with the SEV were performed in reflectance mode, this involved white referencing with a $20 \times 20$ cm SphereOptics Zenith Polymer® SG3120

$\approx$ 99 % full material PTFE diffuse standard to determine the relative reflectance of sample targets. The derived relative reflectance was automatically divided by the calibration values of the white diffuse standard supplied by the manufacturer. An Arrilite Plus 575 W high performance halogen lamp placed at a viewing angle of 45° at a height of 30 cm from sample was used as a light source in a dark calibration laboratory. No additional background corrections were done since the sample was very thick and optically deep (**Figure 1**). Furthermore, the sample had been placed in a dark container with negligible

reflectance. Observations were done 5 cm above sample at nadir making a target pixel with a diameter of 0.7 cm and each measurement was an average of 30 scans. Additional analyses on these measurements were conducted in Mathworks Matlab R2020b including computing the derivative, spectral angle mapping and absorption feature identification as proposed in previous work (Liutkus, 2015;Garaba and Dierssen, 2020;Garaba et al., 2021).

*Insitu* measurements were also completed aboard a small electric motor powered boat on 25 August 2022 on Lake Bante. SEV was operated in radiance mode with each observation set to be an average of 20 scans and pseudo replicate measurements were collected from Station 1 to 11 (**Figure 1a**). Sampling at each station was performed through a series of three radiance measurements over the targets (i) diffuse white panel = $E_d$, (ii) water surface = $L_{sfc}$ and then (iii) sky = $L_{sky}$ at a ~45° viewing



angle from nadir. The diffuse white panel was the same 20 × 20 cm SphereOptics Zenith Polymer® SG3120 ≈ 99 % full material PTFE diffuse standard. Efforts were made to maintain an 90 - 135° azimuthal angle from sensor heading to the sun to mitigate specular reflection (Garaba and Zielinski, 2013). A set of at least 4-6 pseudo replicate observations were achieved at the survey stations as this was dependent on drift and rotation of the boat. Rotation of the boat sometimes resulted in non-optimal viewing geometry that meant presence of surface reflected glint. Spectral reflectance ($R$) was derived by assuming a flat sea surface with a ρ = 0.021 for brevity,

$$R = \frac{L_{sfc} - \rho \cdot L_{sky}}{E_d} \tag{1}$$

## 3 Results

### 3.1 Imaging and visual microscopic properties of the algae

The cells dominating the bloom sample in 2021 were identified as a member of the cyanobacteria genus *Nodularia* that is *N. spumigena* (**Figure 2**). Filaments of *N. spumigena* are unipolar, straight or curved and yellowish, olive-green or blue-green in colour (Guiry and Guiry, 2021). The inherent heterocysts differ in appearance and size only slightly from the vegetative cell. *N. spumigena* is known to inhabit low saline or brackish waters similar to the Lake Bante water type, where it can form larger blooms. One of the first reported cases of *N. spumigena* harmful blooms in Lake Bante dates back to August 1990 (Nehring, 1993). Although not determined in this current study, the 1990 HAB was dense with a 6 cm thickness that was concentrated in the western end of the lake due to the easterly winds experienced typically in August with the temperature range of 15 – 22 °C (Nehring, 1993). By visual inspection, the samples of 2022 were similar to that of 2021.



**Figure 2.** (a-b) FlowCam images highlighting area-based diameter in µm as a measure of particle size and (c-e) Jenoptik ProGres® Gryphax® Kapella microscope photos magnified from 10× to 32× times of *N. spumigena* observed during the bloom in Lake Bante on 16 August 2021.

## 3.2 DNA genetic identification

Full sequences were obtained of the 16S rRNA (Garaba and Bonthond, 2022b) and rbcL genes (Garaba and Bonthond, 2022a) accession number OP925098. Very strong similarities > 99% to *Nodularia spumigena* sequences were obtained from the





BLAST searches against the NCBI nucleotide collection. Both genes had 100% matches with homologues from the strain *N. spumigena* UHCC 0039, of which the genome is available. When we limited the search to type strains only, the type strain of *N. spumigena* (PCC 73104, (Lehtimäki et al., 2000) was identified as the best match with 98.93% and 95.69% similarity to the 16S rRNA and rbcL genes, respectively.

## 3.3 Hyperspectral characteristics

### 3.3.1 Absorption measurements

The absorption coefficient spectra measured with the PSICAM for sampling done in August 2021 and 2022 (**Figure 3**) showed shapes typical for water samples containing phytoplankton. In fact,  it could be seen that the total absorption coefficient spectrum of the water constituents measured was dominated by the phytoplankton pigments, as their absorption peaks were clearly visible and the typical exponential increase towards the shorter wavelengths (Kirk, 2011) caused by CDOM and non-algal particles was less prominent.  This is supported by comparison of the $a_{tot}$ and $a_{cdom}$ measurements (**Figure 3**) in 2022, where it is clearly visible that in most parts of the spectrum $a_{cdom}$ is considerably smaller than $a_{tot}$. The values of $a_{cdom}$ were nearly uniform in the lake, thus spatial changes in $a_{tot}$ were driven by particulate/phytoplankton absorption ($a_{ph}$) Generally, $a_{tot}$ and $a_{ph}$ spectra are dominated by two large peaks in the blue ~440 nm and in the red ~680 nm, attributable to the presence of chlorophyll-a. Additional absorption peaks were revealed by fourth derivatives of the measured absorption coefficient spectra at ~416, 464, 496, 630, 642 and 682 nm (**Figure 3**). The peaks from the derivative analysis are related to chlorophyll-a and the presence of other photosynthetic or photo-protective pigments. The spectral shapes of the collected water samples at the various sampling locations were quite similar, suggesting a general presence of *N. spumigena* in the various areas of Lake Bante. However, a striking difference was the presence of a peak at 574 nm in the non-bloom sample, which was normally present, but missing for both years (2021 and 2022) in the samples with the highest concentration of *N. spumigena*. Also the peaks around 630 nm and 642 nm were absent in the derivative analysis of some sample spectra (**Figure 3**). One possible benefit of knowing about these absorption features would be in simplified algorithms that could be used to optically infer or detect a *Nodularia* bloom for example in Lake Bante using an in-water absorption automated continuous sensor.

In remote sensing, a common way of detecting cyanobacterial blooms involves use of salient spectral features in the measured reflectance or absorption signal caused by the phycobilins pigments unique to the specific phytoplankton group. Four major groups of phycobilins that have different absorption maxima include phycoerythrin (490 – 575 nm), phycoerythrocyanin (570 – 595 nm), phycocyanin (615 – 640 nm), and allophycocyanin (620 – 655 nm) as proposed in literature (Seppälä et al., 2007;Sidler, 1994;Rowan, 1989). Furthermore, satellite remote sensing of cyanobacteria often utilizes the 620 nm waveband as a proxy for phycocyanin (Stefan et al., 2005;Wang et al., 2016). The Lake Bante bloom sample absorption coefficient signal did not reveal this feature although peaks were observed in the derivative spectra at shifted neighbouring wavebands (630 and 642 nm). However, this is not necessarily contradictory to the presence of cyanobacteria, as the phycobilin content in these

organisms can vary largely depending on the physiological status and environmental conditions such as irradiation (Seppälä et al., 2007). It has been reported that photobleaching of phycobilins tend to occur under strong radiation conditions (Donkor and Häder, 1996). As the samples were taken from the surface where ultraviolet and visible irradiation is strongest, we believe this might be related to the absence of peaks in the region of phycocyanin and allophycocyanin. The possible differences in

phycobilin content of cells from various areas of the lake might also be explained by variations in the physiological status of the cells. *Nodularia* blooms usually develop in the upper 5 m layers and float with age towards the surface, where scum formation occurs (Gröndahl, 2009). Therefore, it is likely that the populations of some samples might have had some phycoerythrocyanin while others did not, which could explain also the presence/absence of the observed peak at 574 nm. An alternative explanation, as the peak was absent especially in the very high concentrated samples could be strong masking by

the proportionally higher absorption in the red part of the spectrum in the bloom sample or indicates the presence of other phytoplankton in the non-bloom areas. The presence or absence of the peak at 574 nm could be used in a stepwise binary algorithm to eliminate or identify blooms in remote sensing observations.

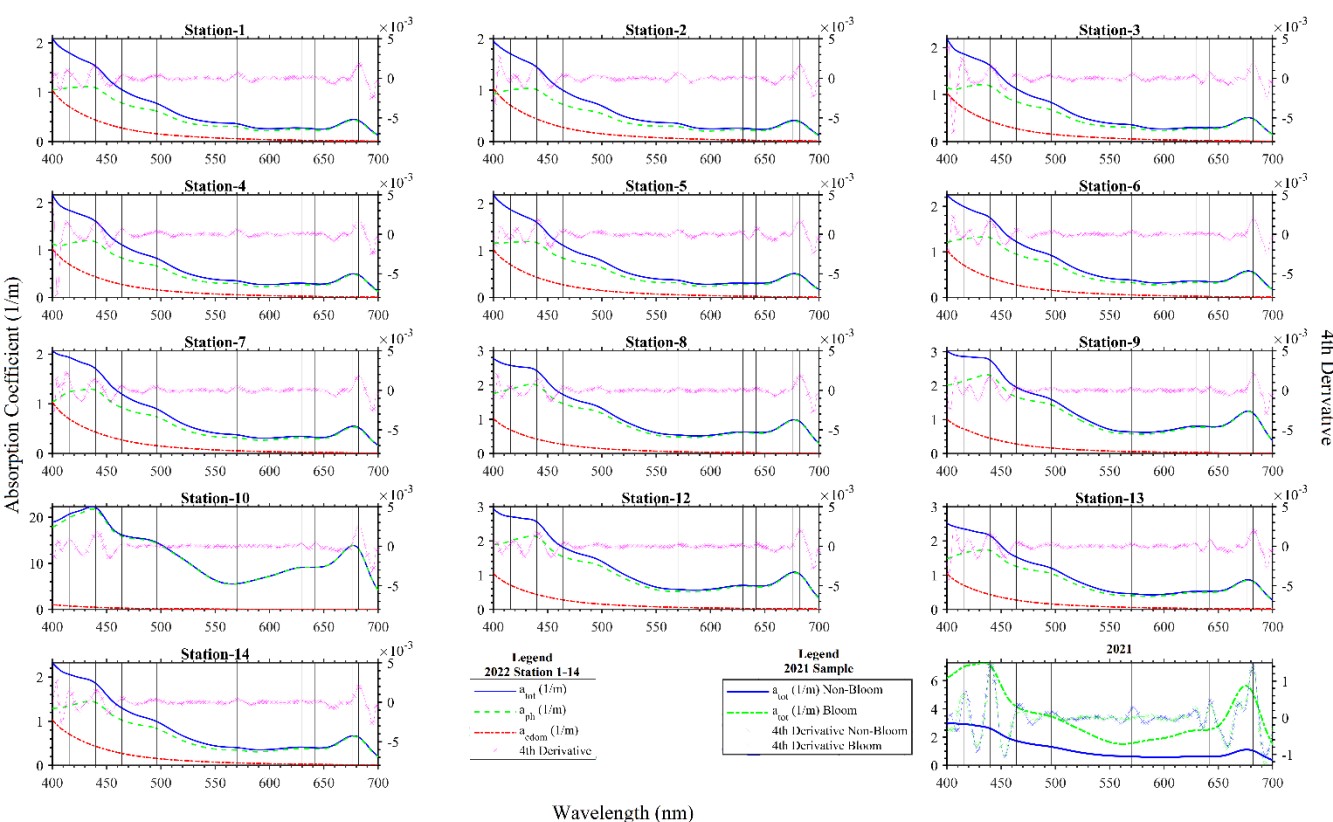

**Figure 3.** Absorption coefficients and 4th derivative spectra for the 25 August 2022 Stations 1-14 and 16 August 2021 sampling of *N. spumigena* bloom events on Lake Bante, Germany. Note Station 11 is missing from the 2022 campaign.

### 3.3.2 Fluorescence Ex/Em

Diagnostic peaks were found in the fluorescent signals of the Lake Bante bloom sample and these are characteristic and common in terrestrial sourced waters (**Table 1**). Three distinct excitation/emission (Ex/Em) regions were revealed in the raw fluorescent spectrum (**Figure 4**).

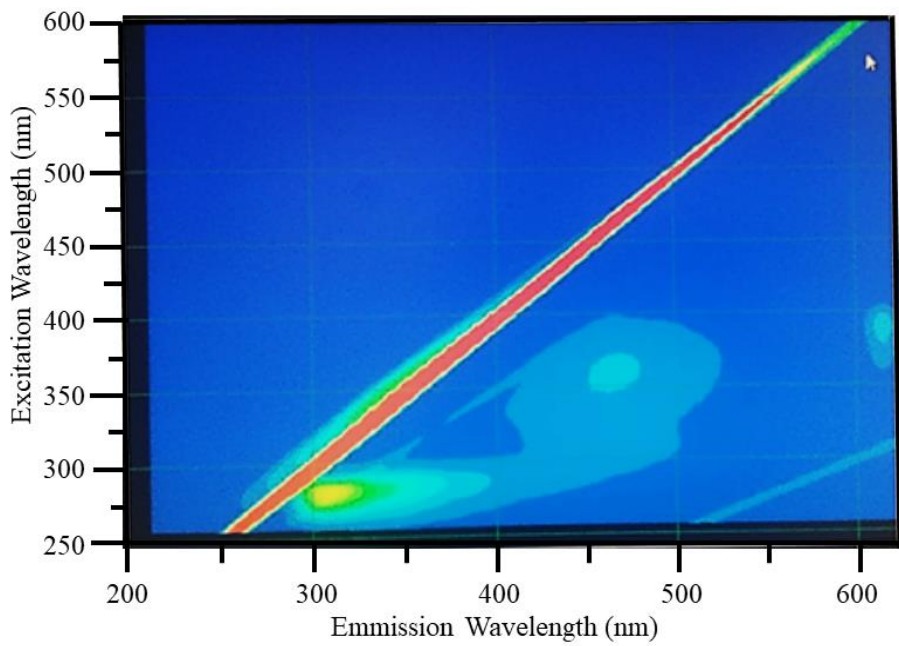

**Figure 4.** Example raw excitation/emission (Ex/Em) fluorescence with diagnostic Ex/Em peaks located at 275/325 nm, 350/475 nm and 390/620 nm were observed in the bloom water from Lake Bante collected on 16 August 2021.

T and B peaks located at Ex/Em = 275/325 nm which indicated a possible presence of autochthonous tryptophan and methionine-like components also associated with biological activity (Coble, 2007;Kwon et al., 2018). Anthropogenic activities around Lake Bante could contribute to the C peak found at Ex/Em = 350/475 nm also related to UVA humic-like compounds of allochthonous origin (Coble, 2007). The lake is surrounded by restaurants, gardens, residential and industrial structure that might contributes to the production as well as the release of humic-like compounds into the water body (**Figure 1**). Pigment-like compounds with the Ex/Em = 390/620 nm have been reported to be associated with protochlorophyllide species (Remelli and Santabarbara, 2018;Myśliwa-Kurdziel et al., 2003;Campbell et al., 1998). These highly fluorescent precursors are present in cyanobacteria exhibiting signatures that can be detected over an emission range of 620 to 650 nm (Böddi et al., 1998). It is also possible that cyanobacteria distributions can be assessed using phycocyanin related peaks instead of chlorophyll-a in vivo fluorescence located in the non-fluorescing photosystem I. Furthermore, a correlation between the filament algae biomass and the intensity of the phycocyanin peak at Ex/Em = 620/650 nm has been proposed as an indicator in remote sensing of algae bloom events (Seppälä et al., 2007).





Example PARAFAC analyses also revealed the presence of 4 main components identified as C1, C2, C3 and C4 (**Figure 5** and **Table 1**). Interestingly, the peak located at 390/620 was missing in our analysis due to a limited sensor sensitivity in the red spectrum. In any case, PARAFAC components derived from our study correspond to well-known chemical groups namely

tyrosine and tryptophane substances linked to C1, C2 components as well as Humic A and Humic C like substances related to the C3, C4 components (Hudson et al., 2007).

**Table 1.** Example descriptive summary of the PARAFAC components derived from bloom water from Lake Bante collected on 16 August 2021.

| Component | Excitation Wavelength (nm) | Emission Wavelength (nm) | Literature | Possible Origin |
|---|---|---|---|---|
| C1 | 278 | 307 | C3 (Wünsch et al., 2015) | *In-situ*, microbiologic |
| C2 | 278 | 348 | C7 (Osburn et al., 2015) | *In-situ*, microbiologic |
| C3 | 258/302 | 465 | C2 (Lin and Guo, 2020) | Allochthonous |
| C4 | 250/376 | 488 | C3 (Gao and Guéguen, 2017) | Allochthonous |
| C5* | 390 | 620 | (Seppälä et al., 2007) | Protochlorophyllide |

*Component could not be derived in PARAFAC but was observed in the raw data by visual inspection.

Fluorescence in our samples was dominated by peaks C1 and C2, commonly referred to as proxies of *in-situ* derived compounds that are indicative of microbiological activity in the investigated waters. The overall composition of the fluorescent spectra shows that up to 30% of the total fluorescence intensity can be assigned to Humic acid like compounds ubiquitous in

coastal waters (Coble, 1996;Repeta, 2015). Similarly, up to 70% of the total fluorescence suggests *in-situ* derived materials namely tyrosine and tryptophane like substances. Enhanced peaks have been associated with the biological reprocessing of DOM in surface waters (Khan et al., 2019).  This is consistent with a bloom algae event in which photodegraded residues of decaying organic matter e.g., cells, exopolymers, or carbohydrates are reutilized by microorganisms in surface waters, therefore producing high values in those indicators (Miranda et al., 2018;Lopes et al., 2020;Weiwei et al., 2019).



**Figure 5.** Four fluorescent components (C1-C4) derived by PARAFAC model for the sampled surface water with *N. spumigena* bloom in Lake Bante on 16 August 2021.

### 3.3.3 Spectral reflectance

As expected in vegetation or algae, a red edge feature was observed in the spectral reflectance measured from the bloom water and in the visible spectrum a peak matching the apparent dark green ~560 nm colour of the bloom was also evident (**Figure 6**). Derivatives analyses showed major absorption features at 298, 437, 633, 676, 837, 986 and 1201 nm. Variability in the pseudo-replicate measurements observed could be related to the drift of the boat during field observations. For the two years, the 2021 data had the highest magnitude reaching ~0.6 whilst the 2022 reflectance was ~0.27 in the near infrared wavebands.



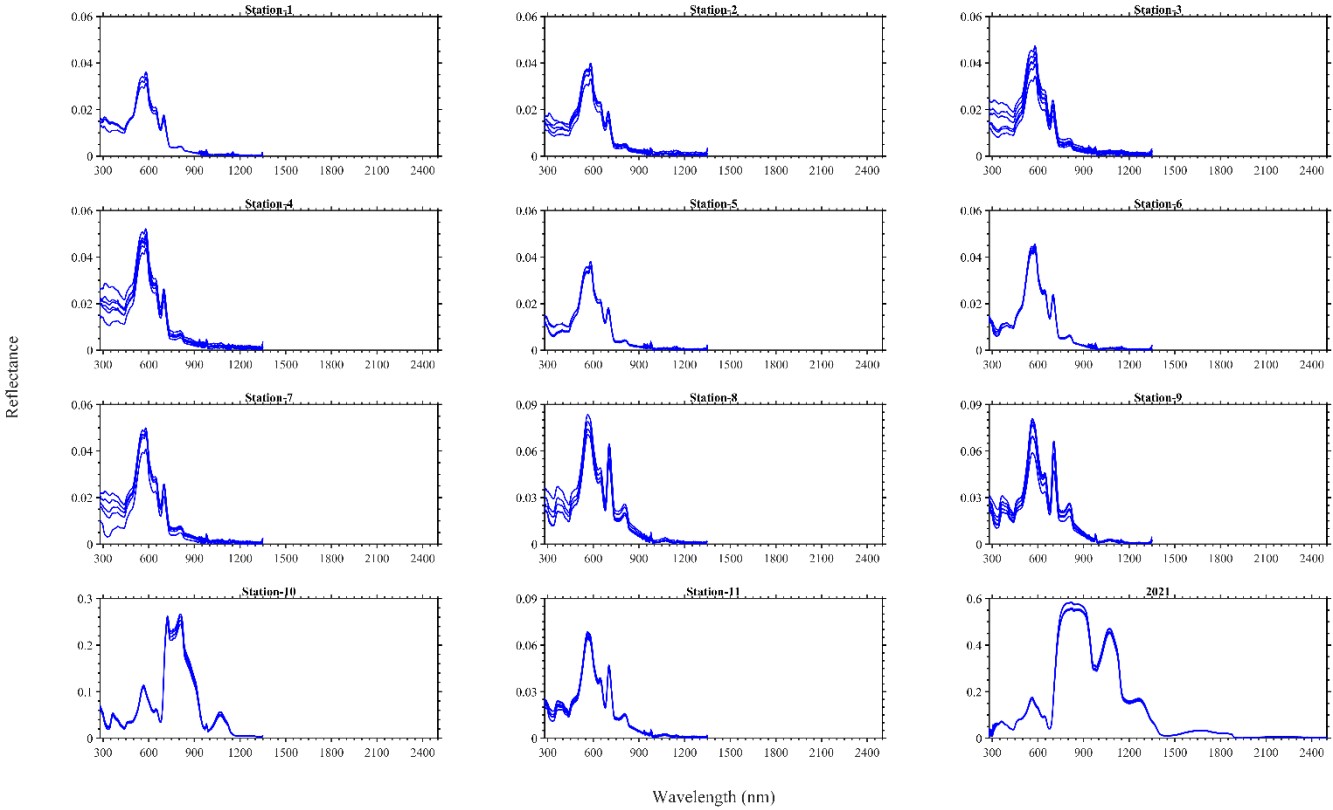

**Figure 6.** Reflectance spectra for the 25 August 2022 Stations 1-11 and 16 August 2021 sampling of *N. spumigena* bloom events on Lake Bante, Germany.

5    We also put the measurement we collected into context by including reported blooms caused by *Ulva prolifera* (Hu et al., 2017), *Tricodesmium* (McKinna et al., 2011) and *N. spumigena* (Soja-Woźniak et al., 2018). A comprehensive quantitative comparison of the spectra was not completed because of the missing metadata in the various studies for example some observations were conducted in a laboratory e.g., (Soja-Woźniak et al., 2018) whilst other observations were on research vessels e.g., (McKinna et al., 2011). However, in all the spectra it is evident they share spectral shape similarities with

10   differences in magnitude (**Figure 7**). Any algorithm based on the measured spectral shape would therefore be appropriate for the detection of such blooms instead of the magnitude dependent approach. A red edge is noticeable in all the spectra and there is diagnostic peak around 550 nm. The *Ulva prolifera* and *N. spumigena* had nearly the same spectral shape over the measured spectrum but reflectance magnitude was alike up to ~750 nm and in the infrared differences were noted.

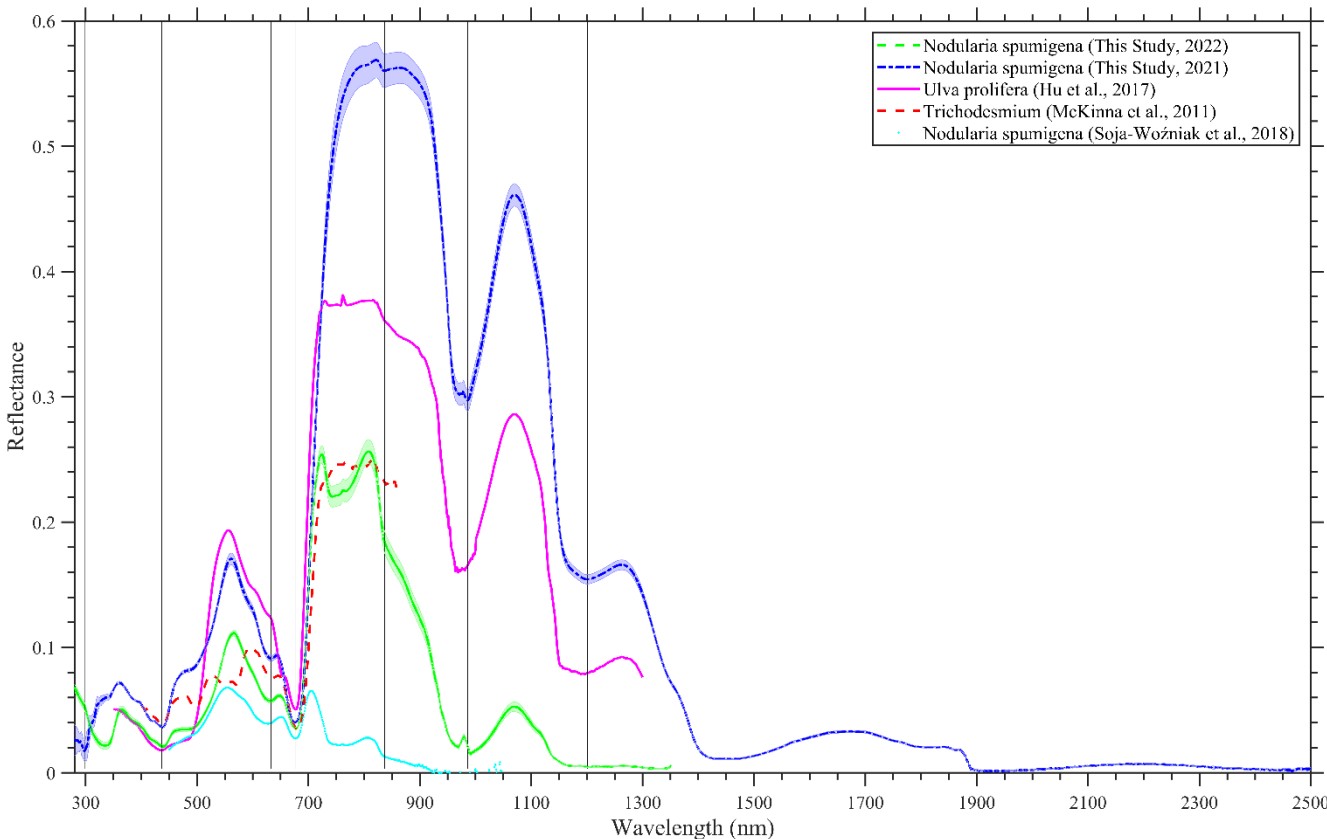

**Figure 7.** A comparison of hyperspectral reflectance spectra of bloom events in Lake Bante, Germany (2021 and 2022), Yellow Sea (Hu et al., 2017), Great Barrier Reef (McKinna et al., 2011) and Baltic Sea (Soja-Woźniak et al., 2018). The Lake Bante spectra are presented as average reflectance and the shading represents 1 standard deviation.

## 3.4 Water quality descriptors

A set of biophysical environmental variables were determined during the 2022 experiment of opportunity. The algal pigments observed were chlorophyll-a and phycocyanin. Average chlorophyll-a concentration was $0.52 \pm 0.37$ mg/L with a range $0.1 - 1.45$ mg/L. However, phycocyanin levels were high variable ranging from 2 to 20.15 µg/L with a mean concentration $5.14 \pm$

10   $5.13$ µg/L. No direct relationship was found between chlorophyll-a and phycocyanin. The water clarity inferred from Secchi disk depth was observed to be indirectly related to the phycocyanin concentration. Overall, water clarity was relatively low $0.77 \pm 0.23$ m with the lowest visibility at 0.2 m and highest 1 m. Salinity in the Lake Bante was generally consistent with a mean $10.57 \pm 0.02$ PSU and ranging between 10.52 to 10.60 PSU suggesting a nearly freshwater environment. Temperature was consistent with the summer season with the surface water expected to be warm. Mean temperature was $23.01 \pm 0.46$ °C

15   whilst the minimum observed was 22.31 °C and highest measurements above 23 °C were around the dense bloom areas (e.g., stations 9-11).



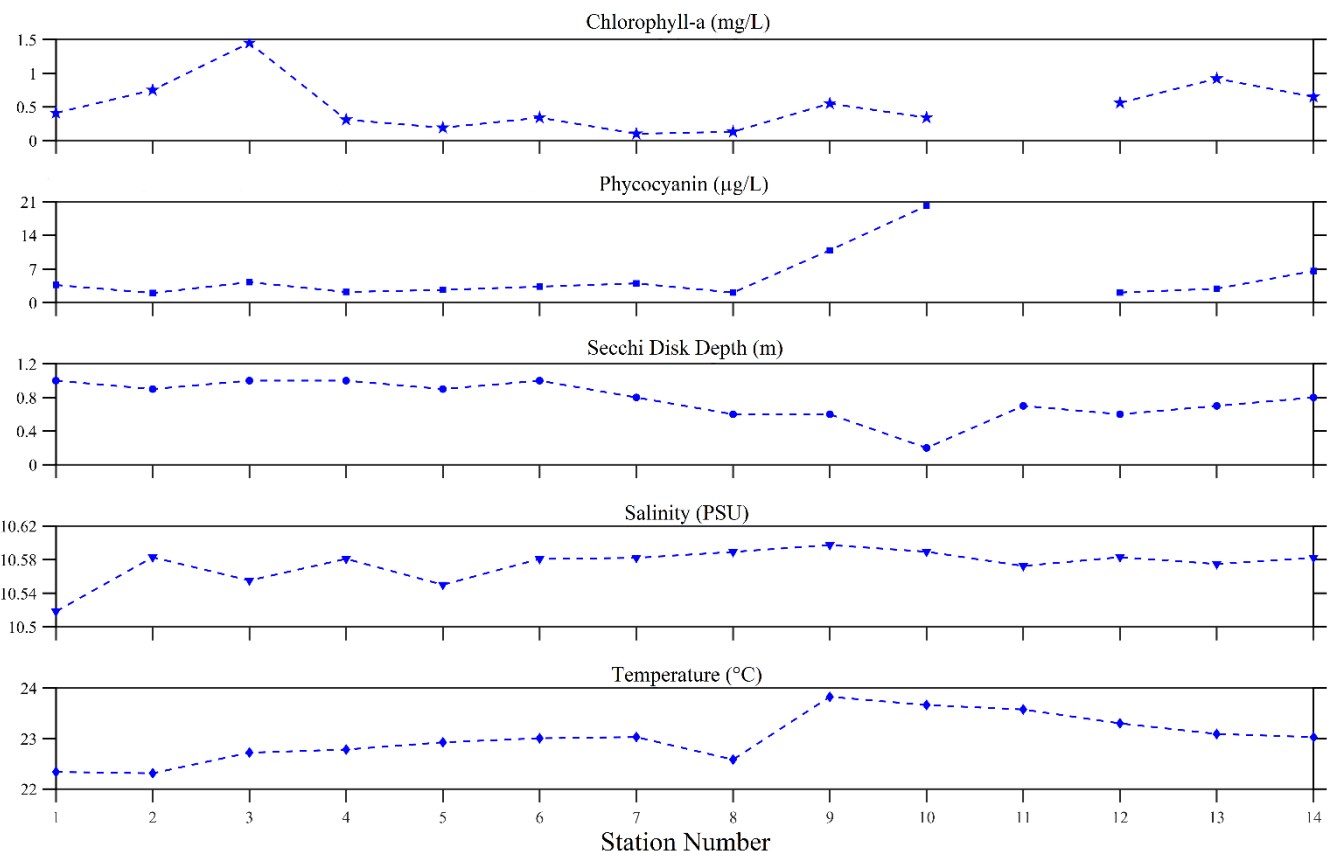

**Figure 8.** Chlorophyll-a, phycocyanin, Secchi disk depth, salinity and temperature from the surface water sampling of *N. spumigena* bloom events on Lake Bante, Germany.

## 4. Discussion, limitations and recommendations

Blooms events linked to *N. spumigena* have been widely reported across the globe e.g., (Kahru and Elmgren, 2014;Leppänen et al., 1995;Mazur and Pliński, 2003;IOCCG, 2021;Olofsson et al., 2020) but hyperspectral characterization of the optical properties of this cyanobacterium is limited. The presented dataset and metadata are therefore expected to contribute to the already available multispectral datasets with the addon benefit of hyperspectral and verified DNA information especially of the identified bloom caused by *N. spumigena*. Despite the limited number of stations studied during our experiments-of-opportunity it is presumed that the gathered high-quality information is a step ahead in advancing scientific evidence-based knowledge especially on the potentially toxic *N. spumigena*.

Of course, without hyperspectral observations and sea-truth measurements it is challenging to identify algae responsible for blooms. We believe a combination of multi to hyperspectral tools as well as auxiliary measurements are a prerequisite in



developing robust diagnostic algorithms for the potential remote identification of *N. spumigena* after successful detection of the bloom. In this study, we do acknowledge the dataset was constrained, thus more efforts are required to continuously monitor changes of various aquatic environments via social media, local newspapers, regular visual inspection, *in-situ* hyperspectral measurements with matching satellite imagery. All these *in-situ* measurements ought to undergo through rigorous curation and
made open-access.

These recommendations apply to the wide range of bloom causing species and water bodies across the globe. Future efforts are thus encourage to further establish endmember open-access databases of comparable and traceable hyperspectral measurements that will be accompanied by essential metadata for robust identification as well as detection of HABs from
remote sensing. Essential open-access metadata would be observations of absorption, scattering coefficients, temperature, salinity, coloured dissolved organic matter, non-algal and algal particle concentrations to better classify a bloom event. Gathering a whole suite of essential variables has the potential to allow a comprehensive identification and characterization of a HAB event, it could also be combined with an optical closure exercise. However, as we inferred from prior related studies e.g. (Dierssen et al., 2015;Seppälä et al., 2007;Kahru et al., 2011), HABs can be challenging to predict. Limitations in
forecasting typically means measurements related to such bloom events tend to be from experiments-of-opportunity whereby metadata or observations of other essential variables are not always readily available due to constraints related to instrumentation, environmental perturbations, platforms or auxiliary tools.

**5 Data availability**

Curation of the observed data was done following the SeaDataNet recommendations. All the datasets are in open-access via
the online repositories. Secchi disk measurements are available at https://doi.pangaea.de/10.1594/PANGAEA.951239 (Garaba and Albinus, 2022). Chlorophyll-a and phycocyanin concentrations are available at https://doi.org/10.4121/21792665.v1 (Rohde et al., 2023). Findings from full sequences are reported on GenBank the 16S rRNA https://www.ncbi.nlm.nih.gov/nuccore/OP918142 (Garaba and Bonthond, 2022b) and rbcL genes https://www.ncbi.nlm.nih.gov/nuccore/OP925098 (Garaba and Bonthond, 2022a). Calibrated spectral radiance measurements
from the 2022 survey are available at https://doi.org/10.4121/21814773.v1 (Garaba and Albinus, 2023) and the reflectance gathered from laboratory observations in 2021 https://doi.org/10.4121/21814977.v1 (Garaba, 2023). Absorption coefficient data is available at https://doi.org/10.4121/21610995.v1 (Wollschläger et al., 2022). Absorbance and flourescence measurements are in open-access at https://doi.org/10.4121/21822051.v1 (Miranda et al., 2023) and https://doi.org/10.4121/21904632.v1 (Miranda and Garaba, 2023).

**6 Conclusions and outlook**

Monitoring of cyanobacteria blooms is of great importance for water quality monitoring and benchmark information can be derived from satellite remote sensing. Although resolving the phytoplankton functional types responsible for a specific bloom



could be limited by the spectral resolution of the satellite sensors, it is feasible to at least detect and map the distributions. In this study we report hyperspectral properties of *N. spumigena* from an event that can be considered a HAB. Our high-quality *in-situ* dataset is therefore expected to help in the algorithm development and potential operational monitoring of similar blooms from current or planned hyperspectral satellite missions e.g., German Aerospace Center - ENMAP, Japan Aerospace

Exploration Agency - HISUI, National Aeronautics and Space Administration - PACE or Italian Space Agency - PRISMA. The current dataset has a unique geographic location, timescale, satellite overpass match-up window ~2 hours as well as hyperspectral measurements which can be considered an addon to related datasets such as the updated European Space Agency Ocean Colour Climate Change Initiative (Valente et al., 2022), National Aeronautics and Space Administration Hyperspectral Imager for the Coastal Ocean reflectance of floating matters (Hu, 2022) or Belgian lakes water quality properties (Castagna et

al., 2022).

The application of the phycocyanin peaks in the fluorescent spectrum associated with the presence of cyanobacteria should be explored in the context of the European Space Agency FLEX mission. Utilizing the diagnostic fluorescent signal, HAB detection and identification could be improved especially when other satellite missions are used in synergy to fuse

hyperspectral and high geo-spatial resolution imagery. Reference measurements would need to combine online surveillance systems with physicochemical parameters providing HAB descriptor conditions such as nitrogen concentration, the intensity of ultraviolet radiation and temperature. It is important to consider the combined use of local knowledge, nowcasting water quality and forecasting of bloom events to potentially focus dedicated interdisciplinary experiments to gather essential as well as comprehensive auxiliary measurements which otherwise tends to be lacking in experiments-of-opportunity.

### Author contribution

SPG conceived the experiments-of-opportunity and prepared the manuscript with input from all co-authors. MA, GB, SF, MLMM, SR, JYLY and JW conducted the laboratory analyses of the samples. MA supported the field campaign. All authors reviewed the text.

**Competing interests**

The authors declare that they have no conflict of interest.

**Funding**

SPG was supported by Deutsche Forschungsgemeinschaft grant no. 417276871 and Discovery Element of the European Space Agency's Basic Activities contract no. 4000132037/20/NL/GLC. PlanetScope imagery was made available through European

Space Agency Third Party Mission Project ID 62280. MLMM was funded by the University of Panama through the Young



Professors Grant number VIP-01-04-16-2018-08 from the Vice rectorate for research and postgraduate studies (VIP) and the National system of research (SNI) from the National secretary of Science and Technology (SENACYT).

**Acknowledgements**

We are grateful to Ulrike Graalmann at the City of Wilhelmshaven, Germany for assisting with permits to conduct research on Lake Bante. Gerrit Behrens, Lutz ter Hell, Helmo Nicolai, Waldemar Siewert and Claudia Thölen were helpful with the preparation and conducting of the field campaign.

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
