# Peer review of "Bio-optical properties of cyanobacterium Nodularia spumigena"

_Earth System Science Data, 2023_

## Author Response (AR1)

**Reply to reviewer**
**Manuscript Title**: Bio-optical properties of cyanobacterium *Nodularia spumigena*
**Authors**       : Garaba et al.
**Journal**        : Earth System Science Data

**Anonymous Reviewer #1**

| Comment | Response | Revision Implemented in Revised Clean Version |
|---|---|---|
| **C1**.
Presented paper shows high quality, comprehensive data set of bio-optical properties of the water with cyanobacteria bloom dominated by *Nodularia spumigena*. Data can be useful in many studies regarding remote monitoring of the *Nodularia spumigena*. Collecting of the data is very well described, giving the user needed overview. Downloading and using the data from the database is easy. | **R1.**
We thank the reviewer for the time taken to review the manuscript and the positive feedback. | Kindly see further revisions as following the suggestions by Reviewer 2. |

**Anonymous Reviewer #2**

| Comment | Response | Revision Implemented in Revised Clean Version |
|---|---|---|
| **C1**.
The authors present a bio-optical data set associated with a harmful algal bloom. The measurements are extensive and should be quite useful for remote sensing and modeling of these blooms, which is a very timely subject. | **R1.**
We appreciate the time taken by the reviewer to provide constructive suggestions to further improve the manuscript. | None. |
| **C2.**
Overall, the manuscript is well written and concise. There are a number of technical issues that need to be addressed, either in a reprocessing of the data or in explanatory text if the processing and corrections were in fact already made. My expertise is in the area of remote sensing reflectance and in-water absorption, and so I will limit my consideration to those topics. I cannot comment on the genetics, microscopy, or fluorescence sections. | **R2.**
Thank you, we have made the specific revisions as suggested below. | See implemented revisions below. |
| **C3.**
Firstly, there is little to no discussion of uncertainty/error on any of the measurements, with the exception of deviation presented with some of the remote sensing reflectance figures (which is useful). | **R3.**
We agree the uncertainty aspect should be discussed in the manuscript. We have added text to further explain the possible sources of error and steps taken during our analyses. | See the revised text **Line 14 Page 17 to Line 6 Page 18**
The *in-situ* sampling and data measurement protocols including error mitigation techniques used during the campaign were considered to be robust as well as modern. Key steps implemented to mitigate these possible uncertainties included (i) before sampling the water containers were pre-rinsed to avoid contamination, (ii) optical observations were an average of 20 - 30 scans that improved signal-to-noise ratio as seen in generated spectra, (iii) modern sampling and analyses protocols were used for the various variables (e.g., radiance, absorbance, absorption, fluorescence, DNA) as well as (iv) rigorous visual inspection of spectra and images. The dynamic nature of the environmental conditions (e.g., wind direction, capillary waves, clouds, currents) during radiance observations on Lake Bante was challenging to avoid but effort was made to guarantee the optimal viewing angles reducing possible surface reflected glint in observations. |

| Comment | Response | Revision Implemented in Revised Clean Version |
|---|---|---|
| **C4.**
In regard to the remote sensing reflectance data - it is not clear to me if the authors accounted for the actual reflectance of the Zenith standard itself - the standard has spectral features, particularly in the UV and SWIR which can substantially alter the resulting calculated water reflectance in these regions if only a single value (eg 99) is utilized. | **R4.**
Thank you for raising this important point.

We have added text to clarify the steps taken during the data analyses. We also further explain that the open-access dataset has the latest calibration reflectance supplied by the manufacturer. | See **Line 9 to Line 11 on Page 8**
Spectral reflectance ($R$) was derived by assuming a flat sea surface with a $\rho = 0.021$ and the manufacturer calibration reflectance of the diffuse white panel was not applied for brevity,

See **Line 7 to Line 9 on Page 19**
Calibrated spectral radiance measurements from the 2022 survey including the manufacturer calibration reflectance of the diffuse white panel are available at https://doi.org/10.4121/21814773.v1 |
| **C5.**
This is especially true if the authors used the spectrometer in "reflectance mode" as mentioned for one of the field campaigns. | **R5.**
We agree that the calibration file must be applied to measurements and we provide this with the dataset. | See **Response R4** and revisions implemented. |
| **C6.**
Secondly, it may be that there is residual skylight that was not removed in the correction of the data which would lead to an increase in blue reflectance< though it is difficult to judge as the data do not appear particularly contaminated - perhaps a comment from the authors would suffice. | **R6.**
Indeed, the blue light can be enhanced, and we believe with the approach applied including visual inspection the correction of 0.021 was appropriate.

We also provide the calibrated measured radiance data to allow future user to apply any surface reflected glint approach. | See **Line 12 to Line 15 on Page 8**
The surface reflected glint as determined from visual inspection was believed to be minimal the Lake was relatively calm during the field campaign. However, to allow future users of the radiance observation to apply surface reflected glint correction of choice the quality controlled calibrated radiometric quantities required (**Equation 1**) were made available in open-access (Garaba and Albinus, 2023). |

| Comment | Response | Revision Implemented in Revised Clean Version |
|---|---|---|
| **C7.**

Also, one of the associated data sets supplies all 3 measurements necessary to derive Rrs using this method, while the other only presents "reflectance" spectra - it is not clear to me that the same set of processing was carried out or if the data are therefore truly comparable. This should be addressed as a source of uncertainty if not. | **R7.**
Maybe this was unclear. In 2021 the measurement was laboratory based in reflectance mode and 2022 this was in-situ in radiance mode. We do acknowledge that comparing such observations can be challenging.

Text has been added to further emphasize this point.

We also add the terms laboratory-based and in-situ measurements to better group the datasets. | See **Line 20 on Page 7**
Laboratory-based relative hyperspectral reflectance measurements of the undiluted sample were conducted on 12 August 2021……..

See **Line 1 on Page 8**
*In-situ* calibrated radiance measurements were also completed aboard a small electric motor-powered boat on 25 August 2022….

See **Line 9 to Line 13 on Page 14**
For the two years, the laboratory-based 2021 data had the highest magnitude reaching ~0.6 whilst the *in-situ* 2022 reflectance was ~0.27 in the near infrared wavebands. Indeed, differences in settings (e.g., density of algae, variable lighting conditions, changes in environment, light source) in the laboratory and *in-situ* spectral measurements could be sources of some uncertainty in the data.

Caption updated
**Figure 7**. A comparison of hyperspectral reflectance spectra of bloom events in Lake Bante, Germany (**laboratory-based** 2021 and *in-situ* 2022), |
| **C8.**

For absorption as measured in the PSICAM - the authors state their assumption that the only (non-water) constituents to a total are a ph and a cdom, and that no absorption due to particulates is present. I agree that these will certainly be the dominant components, however considering the area surrounding the site and its depth, I am not convinced that this is valid, and I would expect some contribution from nonalgal particulates. Can the authors provide evidence for their assumption? | **R8.**
We agree that the assumption needs to be better justified.

We have added text to explain this point further and acknowledge the caveat in our assumption. | See **Line 6 to Line 11 on Page 18**
In terms of the absorption measurements, we acknowledge that there is a caveat with our assumption of the dominant constituents being the dissolved and algae material. During the sampling Lake Bante was relatively calm and most of the surrounding land is either covered by grass, trees, shrubs or paved surfaces suggesting minimal sources as well concentration of non-algal suspended material. Although the absorption coefficient of the non-algal suspended material was not measured in this study, it is suggested that in future research this should be considered to further characterize the optical properties of the water body. |